# Plant-Based HSP90 Inhibitors in Breast Cancer Models: A Systematic Review

**DOI:** 10.3390/ijms25105468

**Published:** 2024-05-17

**Authors:** Ilham Zarguan, Sonia Ghoul, Lamiae Belayachi, Abdelaziz Benjouad

**Affiliations:** 1Center for Research on Health Sciences (CReSS), International Faculty of Medicine, College of Health Sciences, International University of Rabat, Technopolis Parc, Rocade of Rabat-Salé, Sala-Al Jadida 11100, Morocco; lamiae.belayachi@uir.ac.ma (L.B.); abdelaziz.benjouad@uir.ac.ma (A.B.); 2Center for Research on Health Sciences (CReSS), International Faculty of Dental Medicine, College of Health Sciences, International University of Rabat, Technopolis Parc, Rocade of Rabat-Salé, Sala-Al Jadida 11100, Morocco; sonia.ghoul@uir.ac.ma

**Keywords:** plant extracts, bioactive compounds, HSP90 heat-shock proteins, HSP90 inhibitors, breast neoplasms, breast cancer, systematic review

## Abstract

Breast cancer, the most invasive cancer in women globally, necessitates novel treatments due to prevailing limitations of therapeutics. Search of news anticancer targets is more necessary than ever to tackle this pathology. Heat-Shock Protein 90 (HSP90), a chaperone protein, is implicated in breast cancer pathogenesis, rendering it an appealing target. Looking for alternative approach such as Plant-based compounds and natural HSP90 inhibitors offer promising prospects for innovative therapeutic strategies. This study aims to identify plant-based compounds with anticancer effects on breast cancer models and elucidate their mechanism of action in inhibiting the HSP90 protein. A systematic review was conducted and completed in January 2024 and included in vitro, in vivo, and in silico studies that investigated the effectiveness of plant-based HSP90 inhibitors tested on breast cancer models. Eleven studies were included in the review. Six plants and 24 compounds from six different classes were identified and proved to be effective against HSP90 in breast cancer models. The studied plant extracts showed a dose- and time-dependent decrease in cell viability. Variable IC50 values showed antiproliferative effects, with the plant *Tubocapsicum anomalum* demonstrating the lowest value. Withanolides was the most studied class. Fennel, *Trianthema portulacastrum*, and *Spatholobus suberectus* extracts were shown to inhibit tumor growth and angiogenesis and modulate HSP90 expression as well as its cochaperone interactions in breast cancer mouse models. The identified plant extracts and compounds were proven effective against HSP90 in breast cancer models, and this inhibition showed promising effects on breast cancer biology. Collectively, these results urge the need of further studies to better understand the mechanism of action of HSP90 inhibitors using comparable methods for preclinical observations.

## 1. Introduction

Throughout history, natural products derived from plants have played a significant role in cancer therapy and medicine in general. Within the realm of natural products, small molecules synthesized by the plant kingdom have emerged as particularly promising candidates in cancer research. These compounds are often considered “privileged structures”, reflecting their evolutionarily chosen molecular architecture, which affords them enhanced interactions with specific biological targets [1]. The search for anticancer agents from plant sources started in earnest in the 1950s with the discovery and development of the vinca alkaloids, vinblastine and vincristine, and the isolation of the cytotoxic podophyllotoxins. As a result, the United States National Cancer Institute (NCI) initiated an extensive plant collection program in 1960, focusing mainly on temperate regions. This led to the discovery of many novel chemotypes showing a range of cytotoxic activities, including taxanes and camptothecins [2,3]. Notably, approximately one-third of the molecular entities newly approved by the Food and Drug Administration (FDA) were derived from natural products [4,5].

Breast cancer is the most prevalent invasive cancer for women across the globe. It stands as the second primary cause of cancer-related deaths in women [6]. Anatomically, this form of cancer is characterized by an irregular growth of tissues in the breast, originating typically from the inner layers of the milk ducts and glands [7,8]. Histologically, breast cancer is a type of carcinoma because it originates in the epithelial cells of the breast tissue. Current therapy for breast cancer involves a multimodal approach, combining surgery, radiation, chemotherapy, targeted therapy, and hormone therapy, tailored to the specific characteristics of the tumor and the patient. Much progress has also been achieved regarding the combination of these treatments with bioactive phytochemicals from medicinal plants [9,10]. Although these therapeutic interventions have demonstrated significant improvements in survival rates and quality of life, they are not without limitations. The drawbacks of the current therapies include adverse side effects, the development of drug resistance, and an inability to completely eradicate cancer cells in some cases [11]. Therefore, there is an urgent need for developing new effective agents to beat breast cancer more efficiently by reviewing its molecular biology and also by expanding our understanding of the molecular role of bioactive compounds from medicinal plants [12]. From a molecular point of view, breast cancer has been found to be associated with the overexpression of HSP90 [13], a chaperone that interacts with various proteins that promote the development of breast cancer. HSP90 interacts with the estrogen receptor (ER) antiapoptotic kinase Akt, tumor suppressor p53 protein, Raf-1 MAP kinase, angiogenesis transcription factor HIF-1alpha, and receptor tyrosine kinases from the erbB family [14,15]. Consequently, it makes sense to consider the HSP90 pathway in breast cancer therapy [16].

HSP90 is one of the most abundant proteins within eukaryotic cells. It is a chaperone protein that plays a vital role in cell proteostasis, assisting the general folding and stabilization of proteins expressed during cell stress. Therefore, its concentration may increase from 1–2% to 4–6% in response to cellular stress [17,18]. It also targets specific proteins called client proteins that are involved in fundamental cellular processes such as cell growth control, cell cycle progression, signal transduction, gene regulation, and apoptosis [19,20]. However, the list of HSP90-dependent clients is not limited to these processes; more than 400 clients have already been identified, many of which are involved in important biological functions, such as signaling cascades, DNA damage repair, protein trafficking, hormone receptor activation, innate immunity, and many more [20,21,22]. The HSP90 machinery controls the client protein function by accelerating the client’s conformational maturation, which allows for ligand binding and/or the formation of biologically active complexes [23]. Structurally, HSP90 consists of three domains: the N-terminal domain (NTD) that binds ATP, the middle domain (MD) that interacts with client proteins, and the C-terminal domain (CTD) that mediates dimerization [24,25]. Interestingly, HSP90 undergoes conformational changes in response to ATP binding and hydrolysis, cycling between open and closed states (Figure 1) [24]. In humans, there are four major isoforms of HSP90 that differ in their localization, structure, function, and client range. The stress-inducible isoform HSP90α and the constitutively isoform HSP90β are both described in the cytoplasm, while GRP94 and TRAP1 are, respectively, described in the endoplasm and mitochondria of the cell. Additionally, some HSP90s are secreted from the cytoplasm and are commonly called extracellular HSP90 [26]. In cancer, the HSP90 machinery is hijacked to preserve the stability and function of multiple mutated and oncogenic proteins that are essential for the survival and proliferation of cancer cells [21]. This central function puts HSP90 in the position of a potential therapeutic target for cancer treatment [27]. Thus, extensive research is being conducted to develop HSP90 inhibitors for clinical applications. The rationale for targeting HSP90 to treat cancer is based on multiple studies demonstrating that the biology of HSP90 in cancer cells is remarkably different from its basic functions in normal cells. Indeed, HSP90 is highly overexpressed in various cancers [28], complexing with other chaperones, oncogenic proteins, and cochaperones, and exhibiting ~200-fold higher affinity for ATP than homodimeric HSP90 in normal cells [29]. Importantly, the lower affinity shown by the uncomplexed form of HSP90 found in normal cells toward ATP and its competitive inhibitors provides a promising therapeutic window for developing new anticancer agents [19,30]. However, focusing on HSP90 as a therapeutic target in oncology remains a moot point considering that no HSP90 inhibitor has yet been approved for clinical use by the FDA [31,32]. Moreover, it should be noted that the first generation of HSP90 inhibitors inactivates all HSP90 isoforms, which results in a pan-inhibition of HSP90 and detrimental side effects because of the degradation of the entirety of the HSP90 client proteins [33]. The identification of HSP90 isoform-selective inhibitors can be an effective tool for understanding the role played by each isoform in cancer and potentially reducing the toxicities associated with the pan-inhibition. Furthermore, the inhibition of HSP90 through inactivation of its ATPase activity has been shown to induce the so-called heat-shock response, leading to the activation of prosurvival mechanisms that allow cancer cells to escape apoptotic cell death [34,35]. Novel inhibitors that do not induce the heat-shock response could potentially have better clinical applications [36,37].

In the context of cancer therapy, several notable examples include epigallocatechin gallate, gedunin, lentiginosine, celastrol, and deguelin. These molecules exhibited inhibitory activity toward HSP90 [1,16]. In addition to plant-based HSP90 inhibitors, natural products from various sources, such as fungi and bacteria, have been isolated and investigated as potential therapeutic agents. A standout example in this domain is geldanamycin (GA), the first HSP90 inhibitor that was described in the early 1990s by Whitesell and colleagues, who demonstrated HSP90’s crucial role in oncogenic transformation and initiated the concept of chaperone inhibition [32,38]. GA is isolated from the bacteria Streptomyces hygroscopicus; it prevents the ATPase activity of HSP90 by binding to the N-terminal ATP-binding site with high affinity and ultimately impeding cell growth or cell proliferation [16,39]. However, one significant limitation of natural HSP90 inhibitors is their propensity for off-target toxicity, which hinders their direct clinical use. Nevertheless, these natural inhibitors have served as valuable scaffolds for the development of more refined synthetic or semisynthetic HSP90 inhibitors. These modified compounds aim to retain the efficacy of their natural counterparts while minimizing adverse effects, thus paving the way for improved cancer treatments with enhanced safety profiles [5].

In view of all the above, the aim of this systematic review was to identify plant-based compounds and explore their mechanism of action in inhibiting the HSP90 protein and inducing an anticancer effect in breast cancer models. This review focused particularly on in vitro, in vivo, and in silico studies that investigated the biological activity of plant extracts as HSP90 inhibitors in breast cancer models.

## 2. Material and Methods

### 2.1. Search Strategy

The search was conducted in January 2024 utilizing the Boolean equations outlined in Appendix A across four databases: PubMed, Scopus, Web of Science, and Dimensions. Search terms comprised Mesh keywords such as “HSP90 Heat-Shock Proteins” AND “Plant Extracts” AND “Breast Neoplasms” and their synonyms. Additionally, backward snowballing and manual searches were performed to identify relevant papers not captured by the initial search. Only original articles and reviews written in English and published in the past 20 years, from 2003 to March 2024, were considered.

### 2.2. Research Questions

The research question was framed following the PICOS elements (population (P), intervention (I), comparison (C), outcome (O), and study design (S)) as shown in Table 1 below. Therefore, the final research question was as follows: “What is the effectiveness of plant-based HSP90 inhibitors as determined by in vitro, in vivo, and in silico studies in inhibiting breast cancer cell growth through the mechanism of HSP90 inhibition compared to other inhibitors?”

### 2.3. Articles Selection

Reviewers screened titles and abstracts based on predefined eligibility criteria, focusing on studies involving in vitro, in vivo, or in silico assays of plant-based HSP90 inhibitors tested on breast cancer models. The exclusion criteria were (i) studies involving compounds from sources other than plants (e.g., synthesized, fungal, bacterial), (ii) studies working on cancer types different than breast cancer, and (iii) targets different than HSP90 protein. The exclusion criteria are detailed in Figure 2. Disagreements between reviewers were resolved by a third reviewer in a journal club meeting. Full-text articles of selected studies were obtained and evaluated.

### 2.4. Data Extraction

A data extraction form was designed, following the PICOS elements, and each article was analyzed to collect the necessary information on the compounds/plant extracts and cell lines, methods of investigation, comparators used, and main findings of the studies. The three-dimensional structure of HSP90, which has been tested with the plant extracts in the articles, was researched and downloaded in high resolution from the Swiss model in a rainbow color scheme and is used for Figure 3. The “V-shaped” conformation of the mammalian Grp94 homologue from complexes with ADP (PDB ID 2O1V) and full-length human mitochondrial Hsp90 (TRAP1) with AMP-PNP (PDB ID 7KCK) were retrieved from the PDB database. PyMol was used to visualize and prepare protein structures. The HSP90 structures were colored by chain and depicted in ribbons overlayed on the surface in representation at 50% transparency. The NTD was colored in green (residues 1–236), MD in blue (residues 237–617), and CTD in red (residues 618–732). These structures were used to create Figure 1; the HSP90 protein–protein interaction network (PPIN) was generated from the STRING database.

### 2.5. Risk-of-Bias Evaluation

The risk-of-bias assessment criteria and questions were specifically designed based on the PICOS elements of this study and were inspired from the combination of OHAT and QUIN assessment tools (Table 2). The answers were scored from 0 to 3 as follows: adequately specified (score = 3), inadequately specified (score = 2), not specified (score = 1), not applicable (score = 0). For each criterion, the average of the scores for the questions was calculated, and the overall score is the medium of the scores in the questions. Each study was then set into different levels of bias: a score between 2.5 and 3.0 represented a low risk of bias, a score between 1.5 and 2.49 indicated a medium risk of bias, and a score between 0 and 1.49 indicated a high risk of bias.

## 3. Results

### 3.1. Search Results

The conducted search strategy yielded 51 articles across the databases. After eliminating duplicates, the titles and abstracts were screened for relevance (Figure 2). Eleven full-text articles met the inclusion criteria and were included in this study. The articles were published from 2007 to March 2023. In vitro and in silico approaches were described in four articles and one article, respectively, while six studies employed a combination of in vitro with in silico or in vivo approaches.

### 3.2. Risk-of-Bias Evaluation

Different categories of bias were assessed in each study: selection bias, performance bias, detection bias, and reporting bias. Based on the overall score, as shown in Table 3, low and moderate risks of bias were reported, respectively, in five and six papers.

### 3.3. Plant Extracts

Six plants were studied and proved effective against breast cancer models, inducing a decrease in HSP90 expression (Table 4), as follows: the seeds of Foeniculum vulgare, the whole plant of Spatholobus suberectus, the aerial parts of Flueggea leucopyrus, the leaves and stems of Tubocapsicum anomalum, and the leaves of Trianthema portulacastrum and Jasminum multiflorum. Three aqueous [43,45,48]; two methanolic [46,49]; and two ethanolic extracts [42,43] were used. Further extractions and fractioning methods were also used to identify the chemical composition of the plant extracts.

### 3.4. Compounds

A total of 24 compounds were identified. These compounds are presented in Table 5 and categorized into six different classes: eleven withanolides (11:24, 45.8%), six flavonoids (6:24, 25%), three diarylheptanoids (3:24, 12.5%), two phenylethanoids (2:24, 8.3%), one secoiridoid (1:24, 4.1%) and one diterpenoid (1:24, 4.1%). Geldanamycin, Doxorubicin, and Radicicol were used as comparators in the studies.

### 3.5. Techniques Used to Explore the HSP90 Inhibitory Effect of Plant Extracts

The techniques used are summarized and categorized in Table 6. For the in vitro studies, the MTT assay was the predominant technique used to analyze the anticancer effect, and Western blot, RT-PCR, and luciferase-based assays were for HSP90 inhibition. For in vivo studies, immunohistochemistry (IHC) was the most-employed technical approach to localize proteins in tissues. In terms of in silico evaluation, molecular docking was the main method employed to test the compound’s affinity to the HSP90 protein.

### 3.6. Breast Cancer Models

For the in vitro experiments, four cell lines, MCF-7, MDA-MB-231, SKBR3, and BrCSCs, were, respectively, used in seven, five, five, and one articles. For the in vivo experiments, the BALB/c mice model challenged with 4T1 cells was used [48], and female nude mice challenged with MCF-7 and MDA-MB-231 cells [43] and Sprague-Dawley rats with DMBA-induced mammary carcinogenesis were tested [42].

### 3.7. HSP90 Inhibition Pathways Explored In Vitro

Various assays were performed, such as the MTT antiproliferative assay, caspase activity, PARP cleavage assay, JC-1 staining, microscopic fluorescent examination, cell cycle, and DNA fragmentation analysis. The studies reported that compounds such as 2′-hydroxyflavanone, epigallocatechin, withanolides, and oridonin inhibited breast cancer growth and proliferation by inducing apoptosis through various pathways including the mitochondrial pathway [43,47], caspase-dependent pathway [41,44,49], caspase-independent pathway [47], HSP90/HIF-1a cochaperone interaction [43,44,49], and proteasome-dependent degradation of HSP90 client proteins [44,49]. In the MTT assay, the tested compounds exhibited a dose- and time-dependent decrease in cell viability. Western blotting showed a dose- and time-dependent decrease in HSP90 expression in breast cancer cells in all studies, and the induction of HSP70 by withanolides was confirmed by the knockdown of HSP70 using shRNA, enhancing the cytotoxicity in MBA-MB-231 cells [44]. The lowest IC50 value of 1.26 μg/mL (2.7 μM) was observed with the methanolic extract of Tubocapsicum anomalum when tested on the MDA-MB-231 cell line for 24 h [49]. Additionally, an IC50 value of 24.81 µg/mL was observed with Jasminum multiflorum when tested on MCF-7 cells [46]. Lastly, Flueggea leucopyrus (Willd.) exhibited varied concentrations of IC50 depending on the tested cell lines: 27.89 μg/mL for MCF-7, 99.43 μg/mL for MDA-MB-231, and 121.43 μg/mL for SKBR-3 [45], shown in Table 4.

### 3.8. HSP90 Inhibition Pathways Explored In Vivo

The in vivo results showed that female BALB/c mice challenged with 4T1 cells and treated with different doses of fennel extract increased the level of serum glutathione reductase (GR) but could not increase glutathione peroxidase (GPx), the key enzymes in the antioxidant defense system of mice. Fennel extract also inhibited the expression of the Her2 gene and decreased the expression of HSP70 and HSP90 from the ninth day of treatment in all treated groups [48]. Trianthema portulacastrum extract (TPE) downregulated COX-2 and HSP90, blocked IκBα degradation, hampered NF-κB translocation, and upregulated Nrf2 expression and nuclear translocation. TPE treatment also reduced HSP90 expression and increased Nrf2-positive cells when tested on DMBA-induced mammary carcinogenesis in Sprague-Dawley rats [42]. MCF-7 and MDA-MB-231 were used as xenografts to test the oral herbal extracts of the plant extract of Spatholobus suberectus Dunn. The administration of the extract (1 g/kg/d) attenuated the tumor-growth-induced breast cancer xenografts. Epigallocatechin isolated from this extract downregulated LDH-A expression, a feature of cancer cells, and accelerated HIF-1a cochaperone proteasome degradation by interfering with the complex HSP90/HIF-1a [43].

### 3.9. HSP90 Binding Explored In Silico

Different HSP90 models were used for docking analysis in in silico studies, as shown in Figure 3. The N-terminal domain (NTD) of the isoform HSP90α, HSP90 in complex with p50, and HSP90 geldanamycin-binding domain were tested against several compounds [16,40,46]. However, no study used the full-length HSP90 structure. Another study generated a structure-based pharmacophore model of HSP90 and tested it against a dataset of 3210 natural compounds. These compounds were first filtered by drug-likeness parameters and ADMET properties. The resulting 95 druglike compounds were secondly docked into the HSP90 active site and compared with two reference compounds (Geldanamycin and Radicicol). Three hit compounds (Epicalyxins C, Calyxins A, 6-hydroxycalyxin F) showed higher dock scores and more favorable interactions with HSP90 than the reference compounds. The binding stability of these hits was further validated by molecular dynamic simulations [16]. Two mechanisms of action inhibiting HSP90 were described according to the classes of compounds. As shown in Table 5, the first mechanism of inhibition concerned the N-terminal domain (NTD) and was described for the classes of (i) withanolides withaferin A, withaferin A diacetate, and 2,3-dihydrowithaferin A, (ii) flavonoid kaempferol neohesperidoside, (iii) phenylethanoids tyrosol glucoside and 4-hydroxytyrosol, (iv) secoiridoid oleuropein aglycone, and (v) diarylheptanoids epicalyxins C, calyxins A, and 6-hydroxycalyxin F. The second mechanism of inhibition was described, interfering with cochaperones. Tubocapsenolide A, a withanolide, was shown to interfere with the HSP90/HSP70 complex, and the flavonoids epigallocatechin, catechin, gallocatechin, and epicatechin were proven to interfere with the HSP90/HIF-1a complex.

## 4. Discussion

The present review provided a comprehensive overview of the plant extracts and the compounds studied for their HSP90 inhibition effects and explored their binding sites to the protein as well as their mechanism of action inducing an anticancer effect in breast cancer models.

The search strategy of this systematic review followed the guidelines of the PRISMA 2020 statement [50], covering four major databases and using appropriate Mesh keywords and synonyms to capture the most updated literature on the topic. This search strategy resulted in 11 articles that met the inclusion criteria, published from 2007 to 2023. Results were extracted using a data extraction form designed specifically according to the PICOS elements of the review, and the risk of bias was assessed following specific questions for each criterion, combining OHAT and QUIN assessment tools as recommended by other studies [51,52]. The overall quality of the articles ranged from low to moderate, and most of the articles had a low risk of bias for detection bias and reporting bias, indicating that the outcome measures and the results were adequately specified and reported.

Concerning plant extracts, our study showed that (i) at least six plants have been used against breast cancer models, (ii) different parts of the plants were used, and (iii) different types of extracts were proposed. To evaluate the efficacy of these plants, antiproliferation assays were conducted to determine the IC50 value; a lower value indicates a higher efficiency in reducing the cancer cell population. The lowest IC50 described in the studies concerned the methanolic extract of leaves and stems of the Chinese plant Tubocapsicum anomalum. This extract was tested on the MDA-MB-231 cell line and inhibited HSP90 by inducing thiol oxidation and the aggregation of the HSP90-HSP70 complex [49]. In the literature, this plant is described with significant cytotoxic activity and low IC50 values against various cancer cell lines, including breast cancer subtypes [53,54]. Regarding the available data, Tubocapsicum anomalum was the most effective plant against breast cancer cells, with an HSP90 inhibitory effect.

Concerning compounds, the results showed that different classes of plant-based compounds, including diarylheptanoids, diterpenoids, flavonoids, phenylethanoids, secoiridoids, and withanolides, can inhibit the HSP90 activity and modulate its downstream signaling pathways in breast cancer. Withanolides comprised the most studied class [40,44,49]. These data are in accordance with a recent comprehensive review summarizing the efficacy of different classes of naturally derived HSP90 inhibitors in cancerous cell culture and animal tumor models. The review particularly consolidated the primary outcomes in IC50, tumor size, and physicochemical properties of the compounds. In fact, the review also reported that withanolides comprised the most effective class of molecules among the natural HSP90 inhibitors and that the knockdown of HSP70 by shRNA enhanced the cytotoxicity of withanolides in MBA-MB-231 cells [55].

Regarding in vivo studies, fennel, Trianthema portulacastrum, and Spatholobus suberectus extracts were shown to inhibit tumor growth and angiogenesis and modulate HSP90 expression as well as its cochaperone interactions in breast cancer mouse models. These effects are consistent with those described in the current literature on the role of HSP90 in breast cancer biology and therapy. In fact, tumor growth and angiogenesis are reduced when HSP90 is inhibited by KU-32 interaction with the C-terminal domain of HSP90 in trastuzumab-resistant HER2-positive breast cancer cells [56]. These results suggest that HSP90 inhibition is a valuable therapeutic alternative, particularly for breast cancers resistant to chemicals. Another study found that the combination treatment of chemical and HSP90 inhibitors showed (i) synergistic inhibition of the HER2 protein and the downstream PI3K/Akt and Ras/MEK/ERK pathways and (ii) the induction of early apoptotic cell death and G1 arrest in both parent and lapatinib-resistant cells in vitro [57]. These studies suggest that HSP90 inhibitors can not only modulate the expression and activity of different proteins involved in breast cancer progression and resistance but also enhance the efficacy of other anticancer agents.

Concerning in silico assays, different methods such as molecular docking, molecular dynamics simulations, virtual screening, and pharmacophore modeling were reported in the studies [16,40,46] and provided valuable insights into the binding modes, interactions, and mechanisms of action of HSP90 inhibitors. This approach is in line with the current literature on the use of in silico methods for the discovery and optimization of novel HSP90 inhibitors, as discussed in a review where the authors highlighted the current assays and technologies used to find and characterize HSP90 inhibitors such as biophysical, biochemical, and cell-based assays and computational modeling [36]. However, the studies focused on the 3D structure of the N-terminal domain of HSP90 only, while many Homo sapiens full-length structures are available in protein databases, such as Hsp90β in complex with XAP2 and AHR (PDB ID: 8qmo) or Hsp90α in complex with GR and p23 (PDB ID: 7krj), Hsp90α:GR:FKBP52 (PDB ID: 8FFV), Hsp90α:p23 (PDB ID: 7l7j), and finally Hsp90α:Hsp70:HOP:GR (PDB ID: 7kw7). Docking these structures will give better insight into the binding site of the studied compounds as well as the overall understanding of HSP90’s interactions with its client proteins and cochaperones.

The reported compounds were derived from different plant sources and exhibited two mechanisms of inhibition in comparison to the known inhibitors Geldanamycin, Radicicol, and Doxorubicin; the first mechanism was binding to the N-terminal domain (NTD); it is the most-studied mechanism of action and has shown potent in vitro and in vivo efficacy. Indeed, most of the HSP90 inhibitors that have entered clinical trials for cancer treatment target the NTD, which is highly conserved among the isoforms of HSP90 [58]. However, inhibiting the NTD has several limitations, discussed in several reviews [32,55]. NTD-targeting inhibitors can induce the heat-shock response (HSR), which upregulates the expression of heat-shock proteins and can limit the efficacy of HSP90 inhibitors. Additionally, NTD-targeting inhibitors have shown relatively poor pharmacokinetic profiles, which can limit their bioavailability and efficacy in vivo. The NTD of HSP90 is highly conserved across species, which can limit the selectivity of NTD-targeting inhibitors and lead to off-target effects [59]. Furthermore, the NTD of HSP90 is involved in the binding of cochaperones and client proteins, which can limit the specificity of NTD-targeting inhibitors and lead to off-target inhibition [32]. The second mechanism of inhibition used in the reported studies was the disruption of its cochaperone interactions; the withanolide tubocapsenolide A extracted from Tubocapsicum anomalum was reported to inhibit the HSP90/HSP70 chaperone machinery in MCF-7 and MDA-MB-231 cells [44,49], and the flavonoids epicatechin, gallocatechin, catechin, and epigallocatechin extracted from Spatholobus suberectus Dunn inhibited the HSP90/HIF-1a cochaperone interaction when tested in vitro and in vivo against MCF-7 and MDA-MB-231 cell lines and xenograft mouse models [43]. However, targeting the C-terminal domain (CTD) has been poorly studied in breast cancer models, while developing novel HSP90 inhibitors that target the CTD is recommended, since it does not induce the heat-shock response associated with N-terminal inhibitors and it was proven that inhibitors that do not induce heat-shock response could potentially have better clinical applications [32,36,37,60].

An important step toward improving the efficacy of HSP90 inhibitors is the identification and the development of isoform-selective inhibitors of HSP90. The first generation of HSP90 inhibitors inactivates all HSP90 isoforms. This results in a pan-inhibition of HSP90 and detrimental side effects because of the degradation of the entirety of the HSP90 client proteins [33,36]. Therefore, the identification of isoform-selective compounds and inhibitors that target specific domains of HSP90, such as the CTD, can be an effective tool for understanding the role played by each isoform in cancer and potentially reduce toxicities associated with pan-inhibition. However, some challenges and limitations can be identified in the current literature regarding the in silico methodology. First, the accuracy and reliability of in silico methods depends on the quality of the input data, such as the protein structures, ligand conformations, force fields, and scoring functions [61]. Second, the structural diversity of HSP90 inhibitors is limited by the availability of cocrystal structures or pharmacophore models for different domains or binding sites of HSP90 [62]. Third, the translation of in silico findings to experimental or clinical settings requires validation using biochemical, biophysical, or cell-based assays [36].

## 5. Conclusions

This systematic review highlighted the potential of plant extracts and compounds in inhibiting HSP90 activity and inducing anticancer effects in breast cancer models through a rigorous search strategy adhering to PRISMA 2020 guidelines. Results revealed that various plant extracts and classes of compounds, such as withanolides, exhibit promising HSP90 inhibitory effects. Notably, Tubocapsicum anomalum emerged as the most effective plant extract against breast cancer cells, while withanolides demonstrated potent activity among the compound classes studied. In vivo studies further supported these findings, showcasing the inhibitory effects of certain extracts on tumor growth and angiogenesis in breast cancer mouse models. In silico assays provided valuable insights into the binding modes and interactions of HSP90 inhibitors, although they primarily focused on the N-terminal domain. However, targeting the C-terminal and the middle domains remains an underexplored avenue with potential clinical applications.

### Limitations and Future Implications

Despite these advancements, challenges persist: Firstly, the number of in vivo assays is relatively small compared to in vitro and in silico assays. This might restrict the direct application of findings in clinical settings. Secondly, the diverse study designs used among the included studies affect the comparability and reproducibility of results, making it difficult to assess efficacy consistently. To address this, future research should focus on comparable methods and criteria for study design, including animal and cell models, plant sources, extraction methods, dosages, routes of administration, and comparators. Implementing these measures will strengthen the scientific foundation and help find a better candidate for clinal implementations in breast cancer treatment.

## Figures and Tables

**Figure 1 ijms-25-05468-f001:**
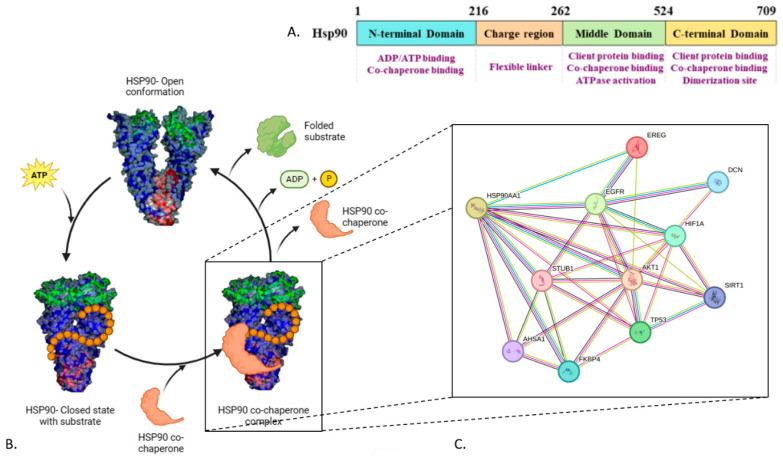
HSP90 structure and its interactions with the oncoproteins involved in breast cancer. (**A**) Schematic representation of HSP90 domains. (**B**) Conformational cycle of HSP90. (**C**) Predicted interactions of breast cancer proteins with HSP90 using network pharmacology (STRING).

**Figure 2 ijms-25-05468-f002:**
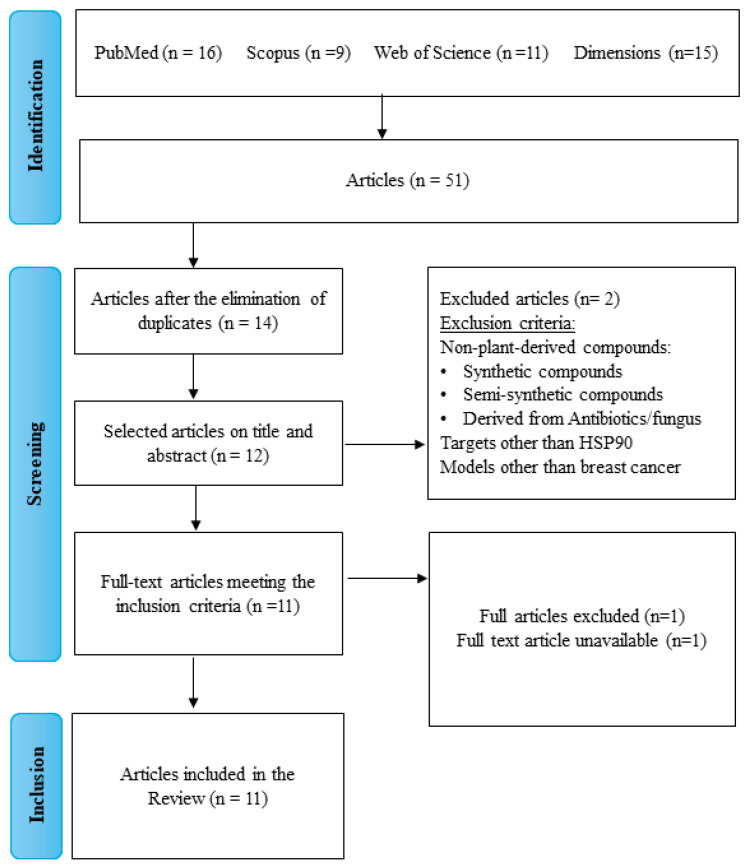
Flowchart of the search strategy.

**Figure 3 ijms-25-05468-f003:**
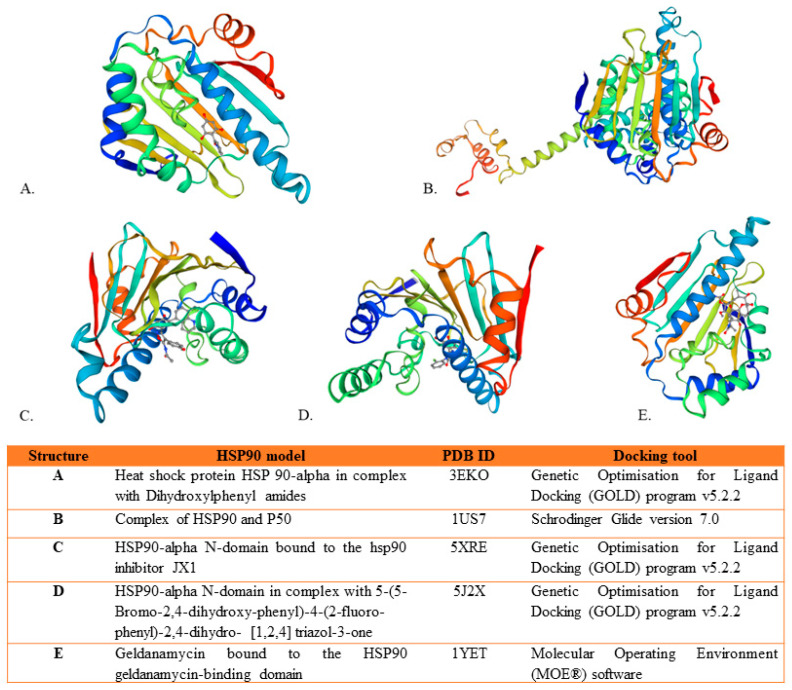
The three-dimensional structures of HSP90 protein used in the included studies. Structures (**A**,**C**,**D**) by [6], (**B**) in [40] and (**E**) was used in study [41]. These structures were retrieved from Swiss model, in rainbow color scheme.

**Table 1 ijms-25-05468-t001:** PICOS elements considered for article inclusion in the review.

PICOS Elements	Criteria
Population	Plant-based Hsp90 compounds
Intervention	Anticancer effect on breast cancer cells and HSP90 inhibition
Comparators	Other HSP90 inhibitors (including nonplant-based HSP90 inhibitors) or no treatment
Outcome	Effectiveness of plant-based HSP90 inhibitors in blocking breast cancer cell growth
Study design	in vitro, in vivo, and in silico studies

**Table 2 ijms-25-05468-t002:** List of criteria and questions that were used to assess risk of bias in the included articles.

Criteria	Questions to Consider
Selection bias	Q1: Did the study clearly state aims/objectives? Q2: Were the plant-based HSP90 compounds chosen based on predefined inclusion and exclusion criteria? Q3: Were the cancer cell lines chosen for the study representative of the type of cancer being studied? Q4: Were the compounds compared to a representative sample of other HSP90 inhibitors?
Performance bias	Q5: Were there any differences in treatment administration between the plant-based HSP90 inhibitors and comparators or control groups? Q6: Were the experimental conditions consistent across all experiments? Q7: For the in silico assays, was the software used to generate computational models validated? Q8: Were the assumptions made in the computational models justified and transparent?
Detection bias	Q9: Were the outcome measures for cancer cell growth and HSP90 protein expression measured using validated and reliable methods? Q10: Were the methods used for measuring these outcomes consistent across all experiments? Q11: Was all of the selected population included in the final analysis? Q12: Were all tested compounds included in the final analysis?
Reporting bias	Q13: Was there any selective reporting of outcomes or results in the study? Q14: Were all outcomes and results reported in a transparent and comprehensive manner? Q15: Were statistical analysis methods appropriate?

**Table 3 ijms-25-05468-t003:** Risk-of-bias evaluation of the included articles.

Articles	Selection Bias	Performance Bias	Detection Bias	Reporting Bias	Overall Score	Quality
Study 1 [42]	🟡 2	🟡 2.75	🟢 3	🟢 3	2.68	Low risk of bias
Study 2 [41]	🟡 2.5	🟡 2	🟡 2.75	🟢 3	2.56	Low risk of bias
Study 3 [43]	🟡 2	⚪0.75	🟢 3	🟢 3	2.18	Moderate risk of bias
Study 4 [44]	🟡 2.5	⚪0.75	🟢 3	🟢 3	2.31	Moderate risk of bias
Study 5 [40]	🟡 2.75	🟢 3	🟡 2.5	🟢 3	2.81	Low risk of bias
Study 6 [6]	🟡 2.25	🟢 3	🔴 1.75	🟡 2.33	2.33	Moderate risk of bias
Study 7 [45]	🔴 1.75	⚪ 0.75	🟢 3	🟢 3	2.12	Moderate risk of bias
Study 8 [46]	🟡 2.25	⚪ 0.75	🟢 3	🟢 3	2.25	Moderate risk of bias
Study 9 [47]	🟢 3	🔴 1.5	🟢 3	🟢 3	2.62	Low risk of bias
Study 10 [48]	🟡 2.25	🔴 1.5	🟢 3	🟢 3	2.43	Moderate risk of bias
Study 11 [49]	🟡 2.5	🔴 1.5	🟢 3	🟢 3	2.5	Low risk of bias

The key: Score = 3: adequately specified (🟢). Score = 2: inadequately specified (🟡). Score = 1: not specified (🔴). Score = 0: not applicable (⚪).

**Table 4 ijms-25-05468-t004:** Plant extracts with HSP90 inhibitory effect on breast cancer models.

Plant Name	Geographical Localization	Part Used	Type of Extract	Study Design	Breast Cancer Model	Treatment Dose	Duration of Treatment	IC50 Value	Outcome	Study
*Foeniculum vulgare*	Iran	Seeds	Aqueous	In vivo	Female BALB/c mice challenged with 4T1 cells	50, 100, and 200 mg/kg; IP injection	Daily, for two weeks	ND	Decreased HSP90 expression	[48]
*Spatholobus suberectus Dunn*	China	All plant	Aqueous ethanol	In vitro	MCF-7, MDA-MB-231	0 μg/mL to 100 μg/mL	Up to 48 h	ND	Inhibits Hsp90/HIF-1a interactions	[43]
				In vivo	Xenograft mouse model	1 g/kg/d, oral intake	Every 3 days for 25 days			
*Flueggea leucopyrus (Willd.)*	Sri Lanka	Aerial parts	Aqueous	In vitro	MCF-7	1, 2, 5, 10, and 20 μM	12 h for Western blot; 48 h for IC50	27.89 μg/mL	Inhibits HSP90	[44]
					MDA-MB-231			99.43 μg/mL		
					SKBR-3			121.43 μg/mL		
*Tubocapsicum anomalum (Solanaceae)*	China	Leaves, Stems	methanol	In vitro	MDA-MB-231	0.1, 1, and 10 μM	24 h and 48 h	2.7 μM (1.26 μg/mL)	Induced thiol oxidation and aggregation of Hsp90-Hsp70	[46]
*Trianthema portulacastrum Linn*	Southeast Asia, tropical America, and Africa	Leaves	Ethanol	In vitro, in vivo	DMBA-induced mammary carcinogenesis in Sprague-Dawley rats	50, 100, and 200 mg/kg in the diet	16 weeks	ND	Decreased HSP90 expression	[45]
*Jasminum multiflorum (Burm. f.) Andrews*	Egypt	Leaves	Methanol	In vitro, in silico	MCF-7	1000–7.81 µg/mL	1 h	24.81 µg/mL	Compounds showed high affinity scores toward HSP90	[41]

**Table 5 ijms-25-05468-t005:** Plant-based HSP90 inhibitors studied in breast cancer models.

Class	Compound	Source	Breast Cancer Model	Mechanism of Inhibition	Method of Investigation	Comparator	Study
Diarylheptanoid	Epicalyxins C	ND	MCF-7, MDA-MB-231	Binds to the NTD of HSP90	Molecular docking	Geldanamycin, Radicicol	[6]
	Calyxins A	ND	MCF-7, MDA-MB-231	Binds to the NTD of HSP90	Molecular docking	Geldanamycin, Radicicol	[6]
	6-hydroxycalyxin F	ND	NA	Binds to the NTD of HSP90	Molecular docking	Geldanamycin, Radicicol	[6]
Diterpenoid	Oridonin	*Rabdosian rubescens*	MCF-7 cell line	ND	Western blot	ND	[49]
Flavonoid	Epicatechin	*Spatholobus suberectus Dunn*	MCF-7, MDA-MB-231 cell lines; xenograft mouse model	HSP90/HIF-1a cochaperone interaction	Western blot, RT-PCR Immunohistochemistry and TUNEL	ND	[43]
	2′-hydroxyflavanone	ND	MCF-7, MDA-MB-231, and SKBR3 cell lines	ND	Western blot, LC-MS/MS	ND	[42]
	Gallocatechin	*Spatholobus suberectus Dunn*	MCF-7, MDA-MB-231 cell lines; xenograft mouse model	HSP90/HIF-1a cochaperone interaction	Western blot, RT-PCR, immunohistochemistry, and TUNEL	ND	[43]
	Kaempferol neohesperidoside	*Jasminum multiflorum (Burm. f.) Andrews*	MCF-7	Binds to the NTD of HSP90	Molecular docking	Geldanamycin	[41]
	Catechin	*Spatholobus suberectus Dunn*	MCF-7, MDA-MB-231 cell lines; xenograft mouse model	HSP90/HIF-1a cochaperone interaction	Western blot, RT-PCR, immunohistochemistry, and TUNEL	ND	[43]
	Epigallocatechin	*Spatholobus suberectus Dunn*	MCF-7, MDA-MB-231 cell lines; xenograft mouse model	HSP90/HIF-1a cochaperone interaction	Western blot, RT-PCR, immunohistochemistry, and TUNEL	ND	[43]
Phenylethanoid	Tyrosol glucoside	*Jasminum multiflorum (Burm. f.) Andrews*	MCF-7	Binds to the NTD of HSP90	Molecular docking	Geldanamycin	[41]
	4-hydroxytyrosol	*Jasminum multiflorum (Burm. f.) Andrews*	MCF-7	Binds to the NTD of HSP90	Molecular docking	Geldanamycin	[41]
Secoiridoid	Oleuropein aglycone	*Jasminum multiflorum (Burm. f.) Andrews*	MCF-7	Binds to the NTD of HSP90	Molecular docking	Geldanamycin	[41]
Withanolide	Withanolide E	*Physalis peruviana*	MCF-7, MDA-MB-231	ND	Western blot, luciferase-based assays, shRNA knockdown	Geldanamycin	[47]
	Tubocapsenolide A	*Tubocapsicum anomalum*	MCF-7, MDA-MB-231	Hsp90/Hsp70 cochaperone interaction	Western blot, luciferase-based assays, shRNA knockdown, and detection of intracellular reactive oxygen species accumulation	Geldanamycin	[46,47]
	Tubocapsenolide B	*Tubocapsicum anomalum*	MCF-7, MDA-MB-231	ND	Western blot, luciferase-based assays, shRNA knockdown	Geldanamycin	[47]
	Tubocapsanolide C	*Tubocapsicum anomalum*	MCF-7, MDA-MB-231	ND	Western blot, luciferase-based assays, shRNA knockdown	Geldanamycin	[47]
	Tubocapsanolide E	*Tubocapsicum anomalum*	MCF-7, MDA-MB-231	ND	Western blot, luciferase-based assays, shRNA knockdown	Geldanamycin	[47]
	Anomanolide A	*Tubocapsicum anomalum*	MCF-7, MDA-MB-231	ND	Western blot, luciferase-based assays, shRNA knockdown	Geldanamycin	[47]
	4b-hydroxywithanolide	*Physalis peruviana*	MCF-7, MDA-MB-231	ND	Western blot, luciferase-based assays, shRNA knockdown	Geldanamycin	[47]
	Peruvianolide H	*Physalis peruviana*	MCF-7, MDA-MB-231	ND	Western blot, luciferase-based assays, shRNA knockdown	Geldanamycin	[47]
	Withaferin A	ND	MCF-7, MDA-MB-231, BrCSCs	Binds to the NTD of HSP90	Western blot, luciferase-based assays, shRNA knockdown, molecular docking	Doxorubicin	[40,47]
	Withaferin A diacetate	ND	MCF-7, BrCSCs	Binds to the NTD of HSP90	Molecular docking	Doxorubicin	[40]
	2,3-dihydrowithaferin A	ND	MCF-7, BrCSCs	Binds to the NTD of HSP90	Molecular docking	Doxorubicin	[40]

**Table 6 ijms-25-05468-t006:** Techniques used and outcomes of the included studies.

Study	Type of Analysis	Technique	Study Type	Outcome
[46]	Anticancer effect analysis	MTT antiproliferative assay	Quantitative	Dose-dependent decrease in cell viability upon TA treatment.
		Caspase-3, -8, and -9 activity assays	Quantitative	Increase in caspase-3, -8, and -9 activities upon TA treatment.
		PARP cleavage assay	Quantitative	Increase in PARP cleavage upon TA treatment.
		Flow cytometry	Quantitative	Cell cycle arrest at G1 phase upon TA treatment.
		Western blotting	Qualitative	Proteasome-dependent degradation of Cdk4, cyclin D1, Raf-1, Akt, and mutant p53, Hsp90 client proteins upon TA treatment.
	HSP90 inhibition effect analysis	Nonreducing SDS-PAGE	Qualitative	Rapid and selective induction of thiol oxidation and aggregation of Hsp90 and Hsp70 upon TA treatment.
		Luciferase refolding assay	Qualitative	Inhibition of the chaperone activity of Hsp90-Hsp70 complex in the luciferase refolding assay.
[42]	Anticancer effect analysis	MTT antiproliferative assay	Quantitative	Decrease in cell viability in all three subtypes of breast cancer cells treated with 2HF.
		Proteomic analysis	Quantitative	Significant changes in the proteins responsible for breast cancer incidence, metastases, and therapeutic sensitivity in breast cancer cells.
	HSP90 inhibition effect analysis	Western blotting	Qualitative	Decrease in HSP90 protein expression in all three subtypes of breast cancer cells treated with 2HF.
[48]	Anticancer effect analysis	Serum GR and GPx measurement by ELISA	Quantitative	Fennel extract increased the level of serum GR in mice.Fennel extract did not increase GPx in all treated groups.
		Immunofluorescence (IFS)	Qualitative	Decreased expression of HSP 70 and 90 proteins in mice.
		Her2 gene expression by QRT-PCR	Quantitative	Fennel extract inhibited the expression of the *Her2* gene in breast cancer.
	HSP90 inhibition effect analysis	Immunohistochemistry (IHC)	Qualitative	Decreased the expression of HSP70 and HSP90 in mice treated with fennel extract.
[41]	Anticancer effect analysis	Neutral red uptake assay	Quantitative	Determined the IC50 = 24.81 µg/mL value of the plant extract.
	HSP90 inhibition effect analysis	Molecular docking	Quantitative	Kaempferol neohesperidoside and oleuropein aglycon showed superior affinity toward HSP90 compared to Geldanamycin.
[40]	Anticancer effect analysis	MTT antiproliferative assay	Quantitative	WFA showed lower IC50 value than that of Doxorubicin.
	HSP90 inhibition effect analysis	Molecular docking	Quantitative	WFA and withaferin A diacetate exhibited strong receptor–ligand interactions against HSP90.
[6]	Anticancer effect analysis	Virtual screening	Quantitative	135 phytochemicals retrieved with satisfying pharmacophore features.
		ADME/T properties analysis	Quantitative	95 natural compounds identified as candidates to inhibit Hsp90.
	HSP90 inhibition effect analysis	Molecular docking	Quantitative	Three compounds identified as better inhibitors than Geldanamycin and Radicicol.
		Pharmacophore modeling	Qualitative	A structure-based pharmacophore model was generated with features complementary to residues required for Hsp90 inhibition.
		Molecular dynamics simulations	Quantitative	The hit compounds retained their intermolecular interactions and position in the binding pocket.
[44]	Anticancer effect analysis	Sulphorhodamine (SRB) assay	Quantitative	Decoction mediates significant cytotoxic effects in all three breast cancer cells phenotypes.
		Fluorescent microscopic examination of apoptosis-related morphological changes	Qualitative	Apoptotic morphological changes observed in all three breast cancer cell lines.
		DNA fragmentation	Qualitative	DNA fragmentation observed in all three breast cancer cell lines.
		Caspase-3/7 assay	Quantitative	Caspase-3/7 were significantly activated in MDA-MB-231 and SKBR-3 cells, indicating caspase-dependent apoptosis in these cells and caspase-independent apoptosis in MCF-7 cells.
	HSP90 inhibition effect analysis	Real-time reverse transcription PCR (RT-PCR)	Quantitative	Inhibition of HSP90 expression mediated by the decoction in MCF-7 and MDA-MB-231, with little effect in the SKBR-3 cells.
		Immunofluorescence analysis of HSP protein expression	Qualitative	No significant effects compared to the controls.
[45]	Anticancer effect analysis/HSP90 inhibition effect analysis	Immunohistochemistry	Qualitative	TPE downregulated COX-2 and HSP90, blocked IκBα degradation, hampered NF-κB translocation, and upregulated Nrf2 expression and nuclear translocation during DMBA mammary carcinogenesis.TPE treatment reduced HSP90 expression and increased Nrf2-positive cells in DMBA-induced mammary tumors in rats.
[43]	Anticancer effect analysis	Apoptosis assay	Quantitative	SS manifested apoptosis-inducing activity in both MCF-7 and MDA-MB-231 cells.
		JC-1 staining	Quantitative	SS activated the mitochondrial pathway apoptosis in breast cancer cells.
		Cell cycle analysis	Quantitative	SS arrested the G2/M checkpoint in breast cancer cells.
		Tumor growth assay	Quantitative	Oral herbal extracts (1 g/kg/d) administration attenuated tumor growth in breast cancer xenografts.
		LDH-A activity assay	Quantitative	SS possessed significant anticancer effects via LDH-A inhibition both in vitro and in vivo.
	HSP90 inhibition effect analysis	Co-immunoprecipitation assay	Quantitative	Epigallocatechin disassociated Hsp90 from HIF-1a.
		Western blot analysis	Quantitative	Epigallocatechin accelerated HIF-1a proteasome degradation.
		Immunohistochemistry assay	Quantitative	Epigallocatechin downregulated HIF-1a expression in breast cancer xenografts.
		LDH-A expression assay	Quantitative	Epigallocatechin downregulated LDH-A expression in breast cancer xenografts.
		Apoptosis assay	Quantitative	Epigallocatechin elevated apoptosis ratio in breast cancer xenografts.
[47]	Anticancer effect analysis	MTT antiproliferative assay	Quantitative	Withanolides reduced cell viability in MDA-MB-231 and MCF-7 cells.
		Apoptosis assay	Qualitative	Withanolides induced cell cycle arrest and apoptosis in MDA-MB-231 and MCF-7 cells.
		Anti-caspase activity analysis	Qualitative	Withanolides induced caspase-3 and PARP cleavage, indicating activation of caspase-dependent apoptosis.
	HSP90 inhibition effect analysis	Western blotting	Qualitative	Withanolides selectively depleted HSP90 client proteins and induced HSP70.
		Luciferase-based assays	Qualitative	Withanolides inhibited HSP90 chaperone activity.
		shRNA knockdown	Qualitative	Knockdown of HSP70 by shRNA enhanced the cytotoxicity of withanolides in MBA-MB-231 cells.
[49]	Anticancer effect analysis	Apoptosis assay	Quantitative	Oridonin induced apoptosis in MCF-7 cells.
		Cell viability assay	Quantitative	Oridonin inhibited cell growth and proliferation in MCF-7 cells.
		Caspase activity assay	Quantitative	Oridonin activated the caspase cascade, leading to apoptosis in MCF-7 cells.
		Observation of morphological changes in cells using phase contrast microscopy	Qualitative	Oridonin induced morphological changes in MCF-7 cells.
		Membrane leakage assay	Qualitative	Oridonin induced membrane leakage in MCF-7 cells.
		Mitochondrial transmembrane potential alternation	Qualitative	Oridonin induced mitochondrial alternations, amplifying the activation of the caspase cascade in MCF-7 cells.
		Calpain-facilitated cell death	Qualitative	Oridonin induced cell death through a caspase-3-independent but caspase-9-dependent pathway in MCF-7 cells.
	HSP90 inhibition effect analysis	Western blot analysis for HSP90 expression	Quantitative	Oridonin downregulated HSP90 expression in MCF-7 cells.

## Data Availability

The data presented in this study are available on request from the corresponding author.

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
