# Peer review of "Plant-Based HSP90 Inhibitors in Breast Cancer Models: A Systematic Review"

_ijms, 2024, doi:10.3390/ijms25105468_

Round 1

Reviewer 1 Report

Comments and Suggestions for Authors

I suggest the abstract be revised to give more methodological details.  

The introduction generally reads well, is informative and provides a context for the study. However, the section could be improved by including the following: (1) Much progress have been achieved  including combination treatments with bioactive phytochemicals from medicinal plants and chemotherapy/radiotherapy in BC (refer to Cancer. 2019 May 15;125(10):1600-1611; Cancers (Basel). 2021 Dec; 13(24): 6222); (2) a strong-enough biological background is needed to review the molecular biology of BC; (3) there is need to expand on the molecular role of other medicinal plant bioactive compounds with the variety of pathways affected in BC (refer to Environ Sci Pollut Res Int. 2022 Apr;29(17):24411-24444;  Plants (Basel). 2024 Jan 5;13(2):153).  

Line 138: Please change “arrested” to “performed” or “conducted”.  

In search strategy, please expand on the following: (1) be more explicit about how each criterion was applied, especially regarding the selection of articles; (2) provide more detail on the search strategies, specifying the Boolean operators and any additional filters used (e.g., AND or OR); and (3) clarify the search terms/keywords and the process of data extraction and how discrepancies were resolved.  

Line 148: Please include a table clarifying the PICOS.  

The inclusion and exclusion criteria for participation in this study are not clearly described in my opinion.  

A period of time encompassed by the search (starting year chose to be included in this review) is not clearly defined.

 All results from tables should be described in much more details in Line 193-217.

 Authors should discuss why plant-based compound has no cytotoxic activity in one study, while killing the same cell in other studies. What does this mean in your opinion?

 In Table 3, different concentrations/doses were used. In your opinion, what is the optimal effective dose in treatment? This should be clearly discussed in the discussion/limitations.

The treatment strategies in Table 3 should be clearly defined. For example, treatment with different culturing media at various doses and incubated for certain time.

In Table 4, the mechanisms of inhibition are unclear. What does it mean by binds to the NTD of HSP90?

 The potential mechanisms of all compounds in BC treatment should be clearly discussed both in vitro and in vivo models in the discussion.

 The review comes to a weak conclusion. Authors should draw conclusions in a separate section with future implications.

 A list of abbreviations should be placed at the end of paper.

Comments on the Quality of English Language

Moderate language editing is needed.

Author Response

Thank you for your time reviewing our paper and your constructive and on-point remarks and suggestions.

Regarding your Comments:

  1. I suggest the abstract be revised to give more methodological details: The abstract was revised; more methodological details were added, we have previously removed most of it because of the word limit of the abstract.
  2. Regarding the introduction: The overall section was revised, (1) was included in lines (52-54) (2) point added in lines (59-60) (3) revised in lines (60-61).
  3. Line 138: Please change “arrested” to “performed” or “conducted”: Arrested changed to conducted in line 143.
  4. Search strategy: the paragraph was improved, detailing the inclusion and exclusion criteria, the time frame the keywords and the Boolean equations used in appendix A. The process of data extraction and how discrepancies were resolved was also mentioned in the paragraph in lines (143-156).
  5. Line 148: Please include a table clarifying the PICOS: The PICOS elements were detailed in lines (158-166)
  6. The inclusion and exclusion criteria for participation in this study are not clearly described in my opinion: The main inclusion criteria of this review were any studies that involve in vitro, in vivo or in silico assays of the effectiveness of plant based Hsp90 inhibitors tested on breast cancer Therefore, any study working on compounds from different sources (e.g. Synthesized, fungus, Bacteria…), or different types of cancer, or different target than hsp90 were excluded. The exclusion criteria are mentioned in the flowchart figure 2. We also have an additional table for the reasons for exclusion of every eliminated article that we can send you upon request.
  7. A period of time encompassed by the search (starting year chose to be included in this review) is not clearly defined: “… the past 20 years were considered…” mentioned in line 149.
  8. All results from the tables should be described in much more details in Line 193-217: The search results were separated and detailed in different paragraphs and tables in the results section.
  9. Authors should discuss why plant-based compound has no cytotoxic activity in one study, while killing the same cell in other studies. What does this mean in your opinion? Our systematic review does not explicitly claim that compounds have different effects; on the contrary, as the techniques used in the studies differed, it was not possible to make direct comparisons. Consequently, any conclusions we draw about a compound's efficacy based solely on IC50 values must be treated with caution.
  10. In Table 3, different concentrations/doses were used. In your opinion, what is the optimal effective dose in treatment? This should be clearly discussed in the discussion/limitations. The optimal effective dose is different from compound to compound, and its efficacy against the target is different from cell line to cell line. Many factors play key roles in determining the effective dose of a compound (specificity, target(s), side effects…) This was discussed in the manuscript as well. However, the review did provide a good understanding of current findings in the field of hsp90 inhibition in cancer therapy, using plant-based compounds, and helped us collect the methodological data we needed for future study directions.
  11. The treatment strategies in Table 3 should be clearly defined. For example, treatment with different culturing media at various doses and incubated for certain time: We are afraid that is not possible since it will complicate the fluency of the table, but we can provide these details as supplementary material upon request.
  12. In Table 4, the mechanisms of inhibition are unclear. What does it mean by binds to the NTD of HSP90? NTD means the N-terminal domain of hsp90 as defined when first mentioned in the introduction line (81) now that the abbreviation list is added in the last page of the manuscript it will make more sense.
  13. The potential mechanisms of all compounds in BC treatment should be clearly discussed both in vitro and in vivo models in the discussion: We have revised the discussion accordingly.
  14. The review comes to a weak conclusion. Authors should draw conclusions in a separate section with future implications: The format of this manuscript was written in accordance with the guidelines of the journal which suggested that the conclusion should be merged with the discussion, each paragraph of the discussion draws its own conclusion and cannot be separated in a section alone because it will be redundant.
  15. list of abbreviations should be placed at the end of paper: A list of abbreviations was added on the last page of manuscript.
  16. Moderate language editing is needed. Thank you for suggesting the language editing, the manuscript has been edited in content and form according to your remarks.

We hope that this new version will be suitable. Thank you for your valuable comments.

Reviewer 2 Report

Comments and Suggestions for Authors

18 March 2024

Ms. Ref. No.: ijms-2928673

Journal: International Journal of Molecular Sciences.

Title: Plant-based HSP90 inhibitors in breast cancer models: A systematic review.

Comments:

Thank you for your efforts in composing an article (A systematic review article) on such a pertinent subject. I have taken the liberty of providing you with a few observations that I believe will serve to enhance the quality of your work. Please find my feedback outlined in the following paragraphs

1-      In the material and methods section of this article mentioned that ‘’the titles and abstracts were evaluated by the reviewers according to the eligibility criteria.’’ Please introduce the eligibility criteria that was used for selecting the references.

2-      There are 58 evaluated references that mentioned in this article and they are between 1970(Reference =34) to 2023 (Reference =35), what was the reason about this timeframe? How was calculated this period?

3-      What was the main inclusion and exclusion criteria for using those valuable references?

4-      There are some different HSPs, why does the authors focusing on HSP90? 

5-      In order to improve the clarity of the introduction, it is recommended that you include some of the following sources as references:

·         https://doi.org/10.3390/ijms25020876

·         https://doi.org/10.1007/s12013-023-01171-y 

Author Response

Thank you for your time reviewing our paper and your on-point remarks and suggestions.

Regarding your Comments:

  1. Search strategy: the paragraph was improved, detailing the inclusion and exclusion criteria, the time frame the keywords and the Boolean equations used in appendix A. The process of data extraction and how discrepancies were resolved was also mentioned in the paragraph in lines (143-156).
  2. Regarding the included articles in the review, the time frame was the papers published in the past 20 years mentioned in line 149. As for the references, the time of publication wasn’t considered, only the relevance to the article with more focus on the most recent papers.
  3. What was the main inclusion and exclusion criteria for using those valuable references? The main inclusion criteria of this review were any studies that involve in vitro, in vivo or in silico assays of the effectiveness of plant based Hsp90 inhibitors tested on breast cancer Therefore, any study working on compounds from different sources (e.g. Synthesized, fungus, Bacteria…), or different types of cancer, or different target than hsp90 were excluded. The exclusion criteria are mentioned in the flowchart figure 2. We also have an additional table for the reasons for exclusion of every eliminated article that we can send you upon request.
  4. There are some different HSPs, why does the authors focusing on HSP90? Indeed, there are different HSPs. The choice of HSP90 specifically came with the rationale that it is the most studied chaperone for cancer therapy in general, and it has more than 400 client proteins that serve as oncogenes. Therefore, it is a very well-established immune checkpoint/target in cancer therapy. We wanted to focus on breast cancer because that’s our lab’s axis of research at the moment.
  5. In order to improve the clarity of the introduction, it is recommended that you include some of the following sources as references: The requested references were included in line 66.

The manuscript has been edited in content and form according to the remarks of the reviewers. We hope that this new version will be suitable. Thank you for your valuable comments.

Round 2

Reviewer 1 Report

Comments and Suggestions for Authors

Dear Authors,

The first-round concerns were not sufficiently addressed. Please make all changes in RED COLOUR.

1. The introduction still needs some work to improve. Please refer to my comments (2) a strong-enough biological background is needed to review the molecular biology of BC; (3) there is need to expand on the molecular role of other medicinal plant bioactive compounds with the variety of pathways affected in BC (please refer to Environ Sci Pollut Res Int. 2022 Apr;29(17):24411-24444;  Plants (Basel). 2024 Jan 5;13(2):153).

2. Line 148-149: A period of time encompassed by the search (starting year chose to be included in this review) should be clearly defined (from....to, not during the past 20 years).

3.  Please include a table clarifying the PICOS. The PICOS description in the text is unclear. Also, the inclusion and exclusion criteria should be clearly described in a SEPARATE SECTION.

4. Line 200-201: Why search is restricted to published articles from 2007 to March 2023? The search should be updated to March 2024. Please make sure that published articles between March 2023 and March 2024 are also included.

5. The treatment strategies in Table 3 should be provided as supplementary material. Treatment with different culturing media at various doses and incubated for certain time should be clarified.

6. I havent seen any changes made on this comment. The potential mechanisms of all compounds in BC treatment should be clearly discussed both in vitro and in vivo models in the discussion.

7. Line 299: The discussion and conclusions should be SEPARATED. There is need to draw conclusions with future implications.

8. The limitations of this REVIEW are MISSING. The limitations with respect to sources of potential bias and imprecision of selected articles are not mentioned.

9. A list of abbreviations should be placed at the end of paper.   

Comments on the Quality of English Language

Minor language editing is required.

Author Response

Dear Reviewer,

Thank you for your thorough review of our systematic review manuscript. We appreciate your insightful comments and suggestions for improvement. Below, we address each of your points and outline the changes we have made accordingly:

  1. Introduction: We have revised the introduction to include a more comprehensive overview of BC molecular pathways, drawing from recent studies such as those you suggested. These modifications have been made and can be found in lines 52-54 and 58-60. Specifically, we have expanded the discussion to provide a stronger biological background on breast cancer molecular biology.
  2. Search Period: The time encompassed by the search was further clarified and highlighted in line 149. Additionally, we have been continuously reviewing the literature using the same research equations and databases in Appendix A to check for any recent publications that weren’t included in the review. We assure you that we are up to date with the literature.
  3. PICOS Table and Criteria: We have included a separate table clarifying the PICOS elements as suggested (Table 1, line 157). Furthermore, we have described the inclusion and exclusion criteria in a distinct section (Article Selection, lines 158-166) for better clarity.
  4. Search Restriction: As mentioned before, we have been keeping an eye on any newly published paper on the subject to be included in the review. We have also updated the search to include articles published up to March 2024 to ensure the inclusion of the most recent literature.
  5. Table 3 and Treatment Strategies: The treatment strategies, including doses and incubation times of the plant extracts, were provided in additional columns in table 3 (now table 4). However, since the cell lines used in these studies are well-established and widely known, along with their corresponding media, we do not see fit to add this information to the manuscript.
  6. Mechanisms of Compounds: In response to your comment, we would like to clarify that all the compounds included in our review share a common mechanism of action, which is the inhibition of HSP90. This shared mechanism was the primary focus of our investigation, and we thoroughly discussed it in the context of both in vitro and in vivo models in the discussion section of the manuscript, emphasizing their implications for breast cancer treatment.
  7. Separation of Discussion and Conclusions: The discussion and conclusion sections were separated (conclusion highlighted in lines 407-418).
  8. Limitations: An additional section on limitations and future implications was included (lines 419-428). We discussed the limitations of our review, particularly regarding potential sources of bias and imprecision in selected articles.
  9. Abbreviations: A list of abbreviations was included at the end of the paper for ease of reference.

Thank you once again for your valuable feedback. We believe that these revisions will strengthen the manuscript and improve its overall quality.

Round 3

Reviewer 1 Report

Comments and Suggestions for Authors

No further comments.